# Pharmacokinetic-Pharmacodynamic Profile, Bioavailability, and Withdrawal Time of Tylosin Tartrate Following a Single Intramuscular Administration in Olive Flounder (*Paralichthys olivaceus*)

**DOI:** 10.3390/ani11082468

**Published:** 2021-08-23

**Authors:** Ji-Hoon Lee, Ga Won Kim, Mun-Gyeong Kwon, Jung Soo Seo

**Affiliations:** Aquatic Disease Control Division, National Fisheries Products Quality Management Service, 216 Gijanghaean-ro, Gijang-eup, Gijang-gun, Busan 46083, Korea; jhlee001@korea.kr (J.-H.L.); gawon1020@korea.kr (G.W.K.); mgkwon@korea.kr (M.-G.K.)

**Keywords:** bioavailability, olive flounder, pharmacokinetics, PK/PD, tylosin tartrate (TT), withdrawal time

## Abstract

**Simple Summary:**

Fishes usually stop eating food when they are sick, and treating diseased fish with oral drugs is a serious hurdle in the aquaculture industry. Tylosin tartrate is a potent bacterial-killing agent useful against frequently occurring bacterial fish infections. We tested the effectiveness against pathogenic bacteria and the human safety of the drug for possible application to cultured olive flounder, one of the most important culture species in far eastern Asian countries. Tylosin tartrate was very effective in killing the pathogenic bacteria grown in artificial culture media, and it was also demonstrated that the drug reached body concentrations in olive flounder, high enough to kill the pathogen. In addition, we also determined how long to wait until the fish clears the injected drug out and it is possible for human consumption. These results will pave a new method for disease treatment useful for olive flounder farming.

**Abstract:**

The objective of this study was to demonstrate the pharmacokinetic–pharmacodynamic profile, bioavailability, and withdrawal time of tylosin tartrate (TT) administered to olive flounder via intramuscular (IM, 10 or 20 mg/kg, *n* = 240) and intravascular (IV, 10 mg/kg, *n* = 90) injections. Serum concentrations of tylosin were determined using a validated liquid chromatography-tandem mass spectrometry method. According to the non-compartmental analysis, the bioavailability of TT was 87%. After the IV injection, the terminal half-life, total body clearance, volume of distribution, and mean residence time of TT were 21.07 h, 0.07 L/kg/h, 2.15 L/kg, and 16.39 h, respectively. Rapid absorption (T_max_ 0.25 h), prolonged action (terminal half-life, 33.96 and 26.04 h; MRT, 43.66 and 33.09 h), and linear dose–response relationship (AUC_0-inf_, 123.55 and 246.05 µg/mL*h) were monitored following 10 and 20 mg/kg IM injection. The withdrawal time of TT from muscle (water temperature, 22 °C) was 9.84 days, rounded up to 10 days (220 degree days). Large C_max_/MIC_90_, AUC_0-inf_/MIC_90_, and T > MIC_90_ values were obtained for *Streptococcus* isolates and these PK/PD indices satisfied the criteria required for efficacy evaluation. This study lays a foundation for the optimal use of TT and provides valuable information for establishing therapeutic regimens.

## 1. Introduction

Antibiotics are typically administered to farmed fish via the oral route. However, the critical disadvantage of oral administration is the difficulty in administering the drug to a sick fish with a poor appetite [1]. Despite the other main disadvantages, such as high labor costs and tissue damage, one of the solutions to the low amount of drug intake is changing the drug from one that allows for oral administration to injection [2]. Olive flounder (*Paralichthys olivaceus*) typically remain calm during injection, so it can be treated without causing excessive stress. A few single and complex antibiotics that could be administered via injection, such as ampicillin, amoxicillin, ceftiofur, and the florfenicol-amoxicillin combination, have been successfully implemented to treat *P. olivaceus* in the Korean aquaculture industry [3].

Tylosin tartrate (TT) is a macrolide antibiotic used in veterinary medicine and is extracted from the soil microbiome called *Streptomyces fradiae* [4]. It is a bacteriostatic antibiotic that binds to the 50S subunit of the bacterial ribosome and inhibits bacterial protein synthesis [5]. TT is known to be strongly effective against Gram-positive and mycoplasma bacteria [6]. To specify, it is broadly used to treat the following conditions: respiratory infection, mastitis, and arthritis in cows; atrophic rhinitis and dysentery in pigs; and mycoplasma infection in birds [4,7,8].

A previous research study examined the efficacy and side effects of TT used to treat *P. olivaceus* infected with *Streptococcus parauberis* [9]. The administration of TT to *P. olivaceus* infected with *S. parauberis* intramuscularly (IM) yielded excellent results: a 90% relative survival rate and no hematological or histopathological side effects. These pharmacological characteristics of TT are beneficial in treating infections caused by Gram-positive bacteria, such as *S. parauberis,* in *P. olivaceus*.

TT distributes extensively in body fluids and tissues because of its high lipid solubility and 40% plasma protein binding [10,11]. The pharmacokinetic profile of TT has been evaluated in various livestock, such as pigs, cattle, chickens, ducks, sheep, and goats [12,13,14,15,16,17,18]. However, there is a lack of information on the pharmacokinetics, bioavailability, and tissue depletion of TT in fish, including *P. olivaceus*.

Moreover, most antimicrobial use in the global aquaculture industry is not related to classification of the target bacteria or susceptibility to the range of available antimicrobials [19], and the used dosage is often based on field results without scientific evidence rather than studies of the antibacterial activity (minimum inhibitory concentration, MIC) of antimicrobials [20]. For successful antimicrobial use in the field, the relationship between the serum levels of tylosin and observed efficacy, as well as the susceptibilities of the causative bacteria (MICs), must be established.

## 2. Materials and Methods

### 2.1. Chemicals

Analytical standard tylosin tartrate (≥98%) was obtained from Dr. Ehrenstorfer GmbH (Augsburg, Germany). All primary analytical reagents for HPLC were purchased from Merck (Whitehouse Station, NJ, USA). All chemicals were of ACS grade purchased from the following sources: sodium phosphate dibasic (Na_2_HPO_4_), sodium dihydrogen phosphate (NaH_2_PO_4_), formic acid, and tricaine methanesulfonate (MS-222) from Sigma (St. Louis, MO, USA). The HLB Oasis column (500 mg, 6 mL) purchased from Waters (Milford, MA, USA) was used for solid-phase extraction (SPE). Brain and heart infusion agar and Muller-Hinton broth were obtained from Difco (Sparks, MD, USA) and lysed horse blood was obtained from Oxoid (Basingstoke, Hampshire, UK). Parenteral TT injection formulation (Tylosin tartrate, injection, 200 mg/mL injectable solution) was purchased from Eeglevet Veterinary Medicine Co., LTD. (Yesan, Chungnam-do, Korea) and diluted with phosphate-buffered physiological saline.

### 2.2. Animals

Healthy *P. olivaceus* weighing 122.9 ± 10.7 g were obtained from a farm located in Pohang, Gyeongsangbukdo. Fish were allowed to adapt to the laboratory conditions for 2 weeks. During this time, they were stored in three circular PVC fish water tanks of 3.0 m (L) × 3.0 m (W) × 1.0 m (H) in size and fed 1% of their body weight with commercial pellet feeds (CJ Feed, Gunsan, Jeonbuk, Korea) per day. Water quality was checked at 09:00 am every day and maintained at approximately 30 PSU, 7–8 mg/L of dissolved oxygen, pH 8.1, and temperature of 22 ± 0.5 °C. All fish were fasted before the experiment to avoid potential adverse effects caused by residual food in their gastrointestinal tract. They were weighed and administered doses based on their body weight. This study followed the protocol approved by the Institutional Animal Care and Use Committee (IACUC, Fish Study Protocol NIFS-2019-3) from the National Institute of Fisheries Science (NIFS). No fish died during the adaptation and sampling period. The absence of tylosin in the muscle and serum was confirmed in five fish before commencement of the experiments.

### 2.3. Experimental Design

The fish were divided into three groups: pharmacokinetic, bioavailability, and tissue depletion group, with 180 (IM dose of 10 and 20 mg/kg, *n* = 90, each group), 90 (IV dose of 10 mg/kg), and 60 (IM dose of 10 mg/kg) fish in each.

Single IM injections of TT were administered into the thick and muscular area under the dorsal fin at 10 and 20 mg/kg to fish maintained at 22 ± 0.5 °C (*n* = 10, each time point) in the pharmacokinetic study. The chosen dose was 10 mg/kg TT because a previous pilot study showed that a single IM injection of TT was effective in combating bacterial infections, such as *S. parauberis* [9]. We also selected double doses (20 mg/kg) to determine if IM administered TT comply with a dose–proportional relationship for the pharmacokinetic parameters, so that it could support extrapolation of the dose levels based on the susceptibility of organisms.

For single IV injection, TT was injected into the caudal vein as a single bolus at 10 mg/kg for bioavailability studies. The location of the needle in the caudal vein was confirmed by aspirating a small amount of blood before injection. Fish injured during the procedure due to excessive bleeding were replaced with new fish.

Ten fish from each group were sampled at 0.25, 0.5, 1, 3, 6, 12, 24, 48, and 72 h after drug administration. To collect blood samples, fish were anesthetized with a light MS-222 aesthetic at a dose of approximately 20 mg/L. Blood samples (1.5 mL) were obtained from the caudal vein using a 1-mL syringe and transferred to a serum separator tube (SST). The blood samples were centrifuged at 3000× *g* for 10 min at 4 °C. Isolated serum was preserved at −80 °C to analyze the drug concentration and determine the pharmacokinetic parameters.

Tissue depletion studies of the withdrawal time were performed in fish maintained as described above. Fish were single-IM injected with 10 mg/kg TT. Muscle samples were obtained 1, 2, 3, 4, 5, and 7 days following the end of the last administration. Muscles were sampled and stored at −80 °C pending analysis for the residue depletion test.

### 2.4. Sample Preparation and HPLC-MS/MS Analysis

The serum or muscle TT concentration was measured using a modified high-performance liquid chromatography-tandem mass spectrometry (HPLC-MS/MS) described in the Korean Food Standards Codex [21]. To simplify, serum or muscle samples (0.5 mL or 2 g, respectively) were homogenized in 6 mL of methanol and centrifuged at 3000× *g* for 10 min for extraction. The clear supernatant was transferred to a 15-mL conical polypropylene tube and the extract was evaporated in a stream of nitrogen at 50 °C. Na_2_HPO_4_-NaH_2_PO_4_ (3 mL, 0.2 M, pH 7.2) was added to the residue. The mixture was loaded onto methanol-water pre-activated (5 mL pre-rinsing) HLB cartridge columns, flushed with 5 mL of 5% aqueous methanol, and eluted with 5 mL of methanol. The eluted extract was evaporated in a stream of nitrogen at 50 °C. The residue was reconstituted with 1 mL of HPLC-grade methanol:water (1:1) and filtered using 0.2-µm syringe filters; 10 µL of the residue were analyzed using the HPLC-MS/MS system.

The HPLC-MS/MS system consists of an Agilent 1260 Infinity series LC (Agilent Technologies, Santa Clara, CA, USA) combined with an Agilent 6430 Triple Quad detector. Chromatography separation was performed using a C_18_ reverse-phase 1.8-µm Agilent Zorbax RRHD SB column (2.1 × 50 mm), which can be maintained at a temperature of 40 °C. During the mobile phase, isocratic elution was performed using the mixture of acetonitrile:water (7:3) containing 0.1% formic acid at a flow rate of 0.25 mL/min. an MS/MS detector was used using the following parameter settings: capillary voltage at 4000 V; nebulizer gas (N_2_); nebulizer gas flow at 11 L/min; nebulizer pressure at 45 psi; gas temperature at 350 °C; and delta EMV^+^ at 500 V. TT was evaluated in multiple-reaction monitoring mode for positive charges. Quantification and qualification ions were *m*/*z* 916.5 to174.1 and from *m*/*z* 916.5 to 101.1, respectively.

The analytical method was validated according to the criteria of the validation procedure [22]. Linearity was evaluated using matrix-matched calibration (MMC) by spiking extracted blanks at four concentration levels (0.5, 1, 10, and 50 ng/g(mL)). The calculated regression lines with standard solutions rendered perfect fits of r^2^ > 0.99. Limit values of detection (LOD) and quantification (LOQ) were determined from signal-to-noise (S/N) ratios of 3:1 and 10:1, respectively. Recovery rates for the extraction procedure from olive flounder tissues were determined by spiking known concentrations of standard solution at three levels (0.5, 5, and 50 ng/g(mL)) into serum or muscle samples. In addition, to avoid the influence of variable instrument sensitivity over time, the linearity of the standard curves for spiked TT standards was checked at the beginning and end to ensure absence of significant fluctuations.

### 2.5. Pharmacokinetic Analysis

The pharmacokinetic parameters for serum concentration were analyzed based on the non-compartmental analysis model using the PKSolver, an add-in program for Microsoft Excel [23]. The peak serum concentration (C_max_) and time to peak serum concentration (T_max_) were determined directly from the experimental data. The area under the concentration–time curve (AUC) was determined by the trapezoidal rule. Bioavailability (*F*) was calculated as *F* (IM) = 100 × (AUC_IM_ x dose_IV_)/(AUC_IV_ × dose_IM_).

### 2.6. In Vitro Antibacterial Activity

In vitro antibacterial activity tests were performed in accordance with the CLSI guideline explained for MIC tests [24]. Bacteria isolated from fish farms in Korea were used for MIC tests, and their detailed strain information is presented in Table 3. Bacteria were incubated at 28 °C in brain-heart infusion agar (BHIA) for 24 h, and their colonies were inoculated to Mueller-Hinton broth (MHB) consisting of 5% horse blood lysates for a 24-h incubation at 28 °C. The test drug TT was serially diluted in 96-well plates consisting of MHB to determine MIC via protocol, described in the VET04-A2. Bacteria were loaded at a density of 5 × 10^5^ colony-forming units (CFU)/mL and cultured at 28 °C for 24 h. The MIC was determined as the lowest antibiotic concentration at which bacterial growth was visibly inhibited. The quality control was performed using *Escherichia coli* ATCC 25922/NCIMB11210; *Aeromonas salmonicida* subsp. ATCC 33658/NCIMB1102; and *Staphylococcus aureus* ATCC 25923 for the precision and accuracy of the test.

### 2.7. Pharmacokinetic (PK)/Pharmacodynamic (PD) Relationships

Surrogate markers for antibacterial efficacy, including the peak serum concentration (C_max_)/MIC, area under the concentration–time curve (AUC)/MIC, and the duration of time (T > MIC) over MIC, were determined using in vitro MIC data and in vivo PK parameters obtained following IM injection of TT at 10 and 20 mg/kg to olive flounder [2,25,26].

### 2.8. Withdrawal Period Calculation

The calculation of the withdrawal time (WT) was performed in accordance with the muscle drug concentration by applying a method developed by the Committee for Veterinary Medicinal Products [27]. The mathematical program used for statistical analysis was WT 1.4 software [28], and the log-linear transformed muscle drug concentration was used as data. Cut-off points were estimated from the upper 95% or 99% at the 95% confidence limit by applying a maximum residue level (MRL) of 0.1 µg/g for tylosin in fin fish muscle [29].

## 3. Results

### 3.1. Analytical Method Validation

An HPLC-MS/MS analysis was used to validate a 0.5 ng/mL limit of quantification used. The retention time of tylosin in the serum was found to be about 0.6 min (Figure 1). The linearity showed a strong linear regression for the MMC (regression line: y = 976.11x − 340.27; correlation coefficient, r^2^ = 0.999) range of 0.5–50 ng/mL. The inter-day and intra-day precision (coefficient of variance) were below 12.8% for three concentrations: 0.5, 5, and 50 ng/mL. The accuracy (recovery rate) ranged from 83.2% to 103.0%, satisfying the criteria of the Bioanalytical Method Validation Guidance for Industry [30]. Details on the parameters are presented in Table 1.

### 3.2. Serum Pharmacokinetics

All fish tolerated TT administered by IM or IV injection well, and no adverse effects were noted. Figure 2 illustrates the semilogarithmic plot of the serum tylosin concentration-time profile following single IM injection at 10 and 20 mg/kg. Tylosin was continuously detected up to 72 h after administration. Table 2 shows the major pharmacokinetic parameters and *F* of tylosin based on the noncompartmental analysis model. After the IV injection, the terminal half-life (t_1/2_λ_z_), total body clearance (Cl), volume of distribution (V_z_), and the mean residence time (MRT) of TT were 21.07 h, 0.07 L/kg/h, 2.15 L/kg, and 16.39 h, respectively. After the IM injection, C_max_ was 10.76 and 16.60 µg/mL, and it took 0.25 h (T_max_) after the administration to reach C_max_, at doses of 10 and 20 mg/kg, respectively. The extent of TT absorption indicated the linear dose–response relationship with AUC following 10 mg/kg IM injection (123.55 µg/mL*h) proportionally increased the 246.05 µg/mL*h at 20 mg/kg. The t_1/2_λ_z_ and MRT of 10 mg/kg IM injection were 33.96 and 43.66 h, respectively. For the 20 mg/kg IM injection, t_1/2_λ_z_ and MRT were 26.04 and 33.09 h, respectively. The TT bioavailability after IM injection was 86.98 to 87.32%. Detailed pharmacokinetic parameters are given in Table 2.

### 3.3. MIC Determination of Clinical Streptococcus Isolates

The MICs of 43 clinical *Streptococcus* isolates are shown in Table 3. Twenty-three *S. iniae* strains showed an MIC range between 0.125 and 0.5 μg/mL. Twenty *S. parauberis* strains showed an MIC range between 0.5 and 2 μg/mL.

### 3.4. PK/PD Relationships

The MICs of TT that restrained 50 and 90% of the clinical *Streptococcus* isolates (MIC_50_ and MIC_90_) are shown in Table 4. Its values were integrated based on the serum concentration in the PK data to determine the C_max_/MIC, AUC/MIC, and T > MIC ratios. According to Table 4, *S. iniae* and *S. parauberis* were calculated with large C_max_/MIC and AUC/MIC values. Moreover, T > MIC was maintained for at least about 2 days, with serum concentrations exceeding the MIC for both strains.

### 3.5. Muscle Withdrawal Time

The residue depletion of tylosin from muscle was monitored after TT single injection at 10 mg/kg, and the results as a function of time are shown in Figure 3. The muscle concentrations sampled 1, 2, 3, 4, 5, and 7 days following a single IM injection at 10 mg/kg were used, where the average concentrations of tylosin were 0.83, 0.66, 0.60, 0.23, 0.22, and 0.08 µg/g, respectively. Withdrawal times were estimated by applying an official MRL of 0.1 µg/g. The calculated withdrawal times of TT were 8.94 days (95% statistical tolerance limit, Figure 3a) or 9.84 days (99% statistical tolerance limit, Figure 3b), which has been rounded up to 10 days (99%).

## 4. Discussion

To the best of the authors’ knowledge, this is the first study describing the pharmacokinetic profile of TT in fish including *P. olivaceus*. Our study provides meaningful pharmacokinetic results of injectable TT in olive flounder, which, in relation to the absorption of TT, is quite rapid from the IM site, showing a long t_1/2_λ_z_ and MRT is beneficial for efficacy, and a linear dose–response relationship with AUC. In addition, by providing a degree day through muscle residue analysis, it was demonstrated that TT is an appropriate drug for treating fish for food.

The dose of IM injection in the pharmacokinetic study was set at 10 mg/kg based on a previous pilot study. Although the pharmacokinetic parameter of IV administration to cultured fish is not important in the application to real practice, it is necessary to calculate the *F* of other routes of administration, such as oral and IM injection. The dose of IV injection was set at 10 mg/kg to calculate factors, such as *F* or V_z_.

The t_1/2_λ_z_ after IV injection was 21.07 h, indicating that the rate of drug elimination is relatively slow in *P. olivaceus*, possibly due to differences between species. Fish have a lower metabolic rate than mammals (t_1/2_ = 4.52 h) [16] and birds (t_1/2_ = 2.04 h) [14], which explains the low elimination rate of TT in *P. olivaceus*. V_z_ represents the volume of distribution to body tissues and fluids, and this was high in *P. olivaceus* (2.15 L/kg). The high V_z_ of TT observed in the present study suggests a high volume of distribution to body tissues of *P. olivaceus*, supported by the results of a previous study in which V_z_ higher than 1 L/kg defined the high volume of distribution of the drug [31]. The extensive distribution of TT throughout the body can be attributed to high lipid solubility and moderate plasma protein binding [14]. Overall, the data suggest that TT has a long half-life and is widely distributed in *P. olivaceus*.

After IM injection at 10 mg/kg, TT reached a C_max_ of 10.76 µg/mL at 0.25 h (T_max_), after being rapidly absorbed into the serum. Although the information about pharmacokinetics of TT after IM injection in other fish may not be applicable to this study, the rapid rate of absorption was similar to the values reported (T_max_ within 1 h) in ducks, cattle, buffaloes, and sheep [14,17,32]. The dose-normalized exposure (AUC) in olive flounder (123.55 µg/mL*h at 10 mg/kg) was 32-fold greater than the one found in the previous study (duck) that used intramuscular administration of the TT (19.14 µg/mL*h after 50 mg/kg) [14], which prolonged the duration of action as evidenced by the large exposure (AUC). The *F* of TT was calculated to be 87.35% in *P. olivaceus* after IM injection at 10 mg/kg, supporting the rapid and nearly complete absorption following IM administration. The comparative pharmacokinetic parameters for the administered dose all showed a fast absorption rate (T_max_, 0.25 h), and when administered at 20 mg/kg rather than 10 mg/kg, the C_max_ was absorbed 1.5 times higher. The AUC was shown to be concentration dependent (123.55 vs. 246.05 at 10 and 20 mg/kg) according to the administered dose.

The calculated withdrawal times by applying an official MRL of 0.1 µg/g [29] were 10 days (99% statistical tolerance limit, rounded up to) when 10 mg/kg of TT was injected at 22 °C. Because of the variability in drug excretion, especially with temperature, a rule of thumb called degree days has been advocated for estimating the required withdrawal time in fish, a poikilothermal animal [33]. It is calculated by adding the mean daily water temperatures (measured in degrees centigrade) for the total number of days measured [1]. Hence, the withdrawal time will be 220 degree days. These data suggest that TT from drug residues will be reduced to ensure reasonable safety for consumers. Unfortunately, there are currently no data on the residual concentrations of TT in all edible fish species, so a sufficient comparison cannot be made.

It is well recognized that macrolides antibiotics as a class exert their activity in parallel with the time length over which tissue concentrations are maintained above the effective level, e.g., MIC [34]. They are so-called ‘time-dependent antibiotics’. Thus, the best therapeutic effects will be expected only when blood levels are sustained for a certain time period by repeated administrations if an oral route is chosen. Achieving the temporary effective concentration is likely to result in a compromised effect in treatment [35]. However, in a previous study, a weak concentration-dependent killing effect of tylosin was reported, unlike the general characteristics of the macrolide antibiotics [14,36]. Temporary effective concentrations can usually guarantee antibacterial efficacy in the case of TT. This unique property will make TT a good candidate for an IM injection route.

The clinical efficacy of antibiotics against their target pathogens can be predicted by PK/PD indices, including C_max_/MIC, T > MIC, and AUC/MIC values. A C_max_/MIC ratio greater than 10 is considered the indicator of activity for concentration-dependent antibiotics [37]. TT, with a partially concentration-dependent effect, was found to satisfy the C_max_/MIC_90_ values for the two test strains (higher than 10.76, Table 4). For time-dependent antibiotics, such as TT, when a threshold concentration of about four times the MIC is reached, T > MIC is a vital pharmacokinetic and pharmacodynamic factor that determines the in vivo effectiveness of TT and needs to be monitored accordingly [34,38]. In addition, AliAbadi and Lees [39] proposed that the maximum serum drug concentration should be at least twice the MIC for pathogenic microorganisms. Our results demonstrated that large T > MIC_90_ values and AUC_0–inf_/MIC_90_ were obtained for *S. iniae* and *S. parauberis* isolates after single IM injection of TT at both 10 and 20 mg/kg to olive flounder (Table 4). The time period with the serum TT concentration exceeding MIC_90_ (T > MIC_90_) obtained for *S. parauberis* lasted for 44 h after single IM injection at 10 mg/kg, and *S. iniae* lasted about 20 more hours (Figure 2 and Table 4). These PK/PD indices support the conclusion that TT is bactericidal against *Streptococcus*.

## 5. Conclusions

This study has shown the beneficial pharmacokinetic profile, including the bioavailability, of an injectable TT formulation in *P. olivaceus* and suggested an estimate of its withdrawal time after single IM administration. In addition, clinical parameters were established by examining the PK/PD relationships using MICs of clinical *Streptococcus* isolates. These data form an important foundation for optimal use of TT as a treatment for systemic infection in *P. olivaceus* and this provides valuable information for establishing scientific and effective treatment regimens.

## Figures and Tables

**Figure 1 animals-11-02468-f001:**
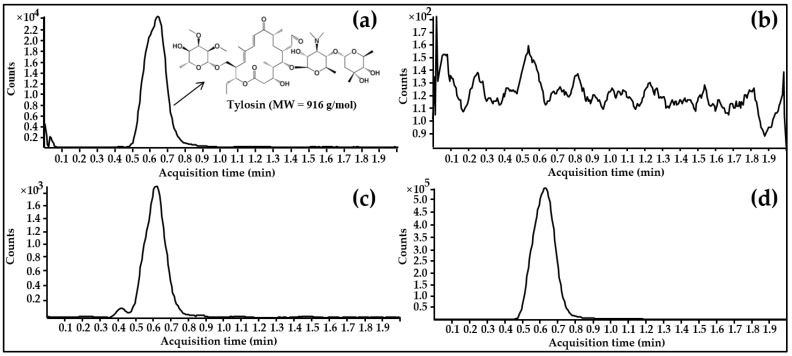
Total ion chromatograms: (**a**) standard solution at 50 ng/mL; (**b**) blank serum sample; (**c**) blank serum sample spiked with tylosin at 10 ng/mL; (**d**) serum sample at 1 h after intramuscular administration of tylosin tartrate.

**Figure 2 animals-11-02468-f002:**
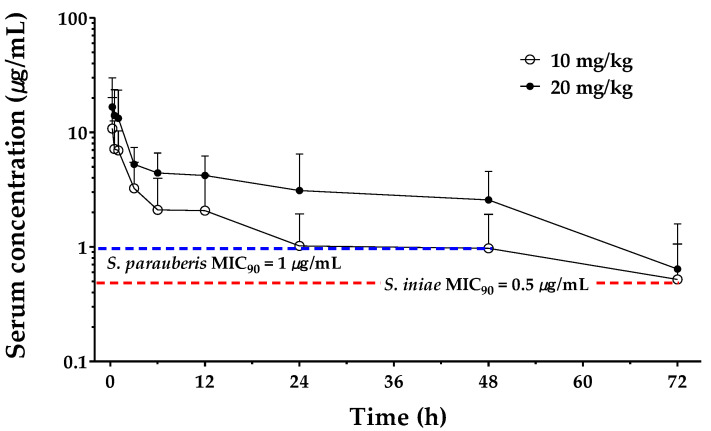
A semilogarithmic plot of the tylosin concentration–time profile in serum following a single intramuscular administration at 10 and 20 mg/kg. Data are expressed as mean ± SD from 10 olive flounders at each time point. The minimum inhibitory concentration (MIC) value corresponds to *Streptococcus parauberis* MIC_90_ (1 µg/mL) and *S. iniae* MIC_90_ (0.5 µg/mL).

**Figure 3 animals-11-02468-f003:**
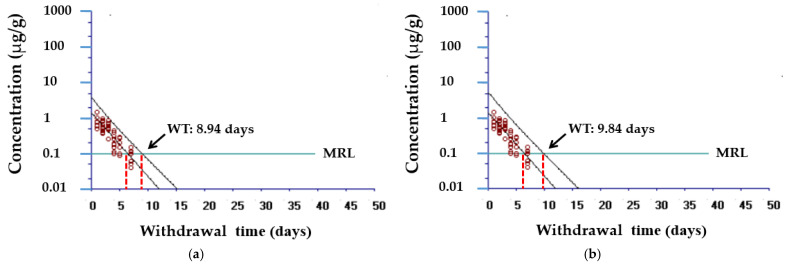
Residue depletion of tylosin from muscle after a single intramuscular administration of 10 mg/kg tylosin tartrate for 1 week: (**a**) 95% statistical tolerance limit with 95% confidence; (**b**) 99% statistical tolerance limit with 95% confidence. Data are expressed from 10 olive flounders at each time point. The maximum residue limit (MRL) used was an official level of 0.1 µg/g.

**Table 1 animals-11-02468-t001:** Validation parameter for the determination of tylosin in olive flounder serum and muscle concentration analysis using HPLC-MS/MS.

Analyte	Matrix	Spike Level (ng/g)	Measured Concentration ^1^ (ng/g)	Intra-Day (*n* = 3)		Inter-Day (*n* = 9)		LOD (ng/g)	LOQ ^3^ (ng/g)
				Accuracy (%)	Precision ^2^ (%)	Accuracy (%)	Precision (%)		
Tylosin	Serum	0.5	0.46	103.0	5.0	89.6	11.7	0.25	0.5
		5	4.21	83.2	2.3	84.5	8.3		
		50	44.20	94.5	3.6	86.4	3.5		
	Muscle	0.5	0.45	93.7	11.6	89.5	12.8	0.25	0.5
		5	4.58	89.5	8.8	92.3	9.0		
		50	45.28	91.6	8.4	90.2	9.3		

^1^ Data are expressed as mean from 12 samples. ^2^ Precision (relative standard deviations, RSDs) must be <20% compliance with the European Commission [22]. ^3^ LOQs were lower than the reported MRLs set by the Committee for Veterinary Medicinal Products [29] for fin fish muscle.

**Table 2 animals-11-02468-t002:** Pharmacokinetic parameters of tylosin tartrate following a single intramuscular and intravenous administration to olive flounder.

Parameter	Unit	Single Dose of Tylosin Tartrate		
		IV 10 mg/kg	IM 10 mg/kg	IM 20 mg/kg
λ_z_	1/h	0.03	0.02	0.03
t_1/2_λ_z_	h	21.07	33.96	26.04
T_max_	h	NA	0.25	0.25
C_max_	µg/mL	NA	10.76	16.60
AUC_0-t_	µg/mL*h	133.63	98.20	221.83
AUC_0-inf_	µg/mL*h	141.44	123.55	246.05
AUMC_0-inf_	µg/mL*h^2^	2318.13	5393.46	8142.39
MRT	h	16.39	43.66	33.09
V_z_	L/kg	2.15	-	-
Cl	L/kg/h	0.07	-	-
*F*	%	-	87.35	86.98

Data are expressed as the mean from 10 olive flounders. λ_z_, first-order rate constant associated with the terminal portion of the curve; t_1/2_λ_z_, terminal half-life; T_max_, time to peak serum concentration; C_max_, peak serum concentration; AUC_0-t_, area under the concentration–time curve from time zero to time t; AUC_0-inf_, area under the concentration–time curve from zero to time infinity; AUMC, area under the first moment curve; MRT, mean residence time; V_z_, apparent volume of distribution; Cl, total body clearance; *F*, systemic bioavailability; NA, not applicable; IV, intravenous; IM, intramuscular.

**Table 3 animals-11-02468-t003:** In vitro antibacterial activity minimum inhibitory concentrations (MICs) tylosin tartrate against *S. iniae* and *S. parauberis*.

Bacterium Name	Isolation Sources (Number of Strains)	Strain Codes	MIC (µg/mL)
*Streptococcus iniae*	Busan, olive flounder, 2004 (*n* = 1)	FP2140	0.5
	Jeju, olive flounder, 2004 (*n* = 2)	FP2149	0.125
		FP2150	0.25
	Ulsan, olive flounder, 2004 (*n* = 1)	FP3060	0.25
	Tongyeong, rock bream, 2006 (*n* = 1)	FP3187	0.25
	Pohang, olive flounder, 2007 (*n* = 1)	FP3358	0.125
	Taean, rock fish, 2008 (*n* = 1)	FP3476	0.25
	Jeju, olive flounder, 2004 (*n* = 4)	FP4033	0.25
		FP4143	0.5
		FP4160	0.125
		FP4164	0.25
	Wando, olive flounder, 2004 (*n* = 1)	FP4080	0.25
	Ulsan, olive flounder, 2005 (*n* = 1)	FP5162	0.25
	Jeju, olive flounder, 2006 (*n* = 1)	FP6085	0.125
	Tongyeong, rock fish, 2012 (*n* = 1)	FPa4413	0.125
	Tongyeong, olive flounder, 1998 (*n* = 1)	BS9	0.25
	Yeosu, rainbow fish, 2010 (*n* = 1)	RaB6-1-a	0.25
	Yeosu, saddled weever, 2010 (*n* = 1)	SW9-2-a-an	0.5
	Yeosu, stripey, 2010 (*n* = 1)	st11-1-b-an	0.5
	Gyeongsangbukdo, olive flounder, 2003 (*n* = 4)	A11022	0.125
		A11024, A11025	0.25
		A11023	0.5
*Streptococcus parauberis*	Jeju, olive flounder, 2003 (*n* = 2)	KSP1	0.5
		KSP4	1
	Jeju, olive flounder, 2004 *(n* = 3)	KSP5, KSP10	0.5
		KSP6	1
	Jeju, olive flounder, 2005 (*n* = 2)	KSP14, KSP20	1
	Haenam, olive flounder, 2005 (*n* = 1)	KSP22	2
	Wando, olive flounder, 2005 *(n* = 2)	KSP40	1
		KSP24	2
	Jeju, olive flounder, 1999 (*n* = 1)	KSP45	1
	Jeju, olive flounder, 2018 (*n* = 9)	SPOF18J1, SPOF18J3, SPOF18J4, SPOF18J5, SPOF18J6, SPOF18J7, SPOF18J9, SPOF18J10, SPOF18J11	1

**Table 4 animals-11-02468-t004:** Pharmacokinetic-pharmacodynamic (PK/PD) integration of tylosin tartrate based on pharmacokinetic data in olive flounder and in vitro MICs.

Bacterial Strain ^1^	*S. iniae*	*S. parauberis*	*S. iniae*	*S. parauberis*
MIC ^2^ (µg/mL)				
Range	0.125–0.5	0.5–2	0.125–0.5	0.5–2
MIC_50_	0.25	1	0.25	1
MIC_90_	0.5	1	0.5	1
TT ^3^ doses (mg/kg, IM)	10 mg/kg		20 mg/kg	
C_max_/MIC_50_	43.04	10.76	66.40	16.60
C_max_/MIC_90_	21.52	10.76	33.20	16.60
AUC_0-t_/MIC_50_ (h)	392.80	98.20	887.32	221.83
AUC_0-t_/MIC_90_ (h)	196.40	98.20	443.66	221.83
AUC_0-inf_/MIC_50_ (h)	494.20	123.55	984.20	246.05
AUC_0-inf_/MIC_90_ (h)	247.10	123.55	492.10	246.05
T > MIC_50_ (h)	84.36	44.75	101.71	62.66
T > MIC_90_ (h)	64.56	44.75	81.92	62.66

^1^ Twenty-three test strains of *Streptococcus iniae* and 20 test strains *S. parauberis*. ^2^ MIC, minimum inhibitory concentration. ^3^ TT, tylosin tartrate.

## Data Availability

All data are contained within the article. Please contact first author and the corresponding author for additional data requests.

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
