# Peer review of "Pharmacokinetic-Pharmacodynamic Profile, Bioavailability, and Withdrawal Time of Tylosin Tartrate Following a Single Intramuscular Administration in Olive Flounder (Paralichthys olivaceus)"

_animals, 2021, doi:10.3390/ani11082468_

Round 1

Reviewer 1 Report

The present paper describes the pharmacokinetics, bioavailability, and withdrawal time of tylosin tartrate (TT) in olive flounder fish. Overall, the manuscript is well written, experiments were properly conducted, executed, and presented. Pharmacokinetics of Tylosin tartrate is well established and studied by various researchers in variety of animals like chicken, duck, pigs, sheep etc but no or limited data available in fish. So, I believe that it is important to study Tylosin tartrate pharmacokinetics in fish, which is helpful for aqua industry. Also, studying withdrawal time/tissue depletion of the drug is very important for safety  of humans.

I have few minor comments and  request authors to address them.

Experimental design: Page 3, line 103-110. It was not clear how many fish in each group (Pk, Bioavailability and Tissue depletion) and also for 10 mg/ kg dose and 20mg/kg dose. I request please re-write this para for better understanding.

2.4. Sample preparation and HPLS-MS/MS analysis: please correct HPLS as HPLC.

Section 2.4. How absolute quantification of drug performed in serum and tissues, did authors used internal standard method or external standard method. In Bioanalysis, internal standard methods are widely used and acceptable to compensate the sample predation and analytical errors. Please provide the details of internal standard used (if internal standard method used).

Page 4; line 153: How many standards were included in the curve. I understand its only 3 standards (correct if I am wrong).

Table 1: Please include column with calculate concentration.

Discussion: please discuss about the pk values vs dose (10 mg/kg and 20 mg/kg) (dose propionate etc).

Author Response

Response to Reviewer 1 Comments

Point 1: The present paper describes the pharmacokinetics, bioavailability, and withdrawal time of tylosin tartrate (TT) in olive flounder fish. Overall, the manuscript is well written, experiments were properly conducted, executed, and presented. Pharmacokinetics of Tylosin tartrate is well established and studied by various researchers in variety of animals like chicken, duck, pigs, sheep etc but no or limited data available in fish. So, I believe that it is important to study Tylosin tartrate pharmacokinetics in fish, which is helpful for aqua industry. Also, studying withdrawal time/tissue depletion of the drug is very important for safety of humans.

Response 1: The authors appreciate valuable comments from the reviewer.

Point 2: I have few minor comments and request authors to address them.

Response 2: We tried to improve the manuscript in accordance with the reviewer’ comments. More information is added in appropriate sections and highlighted. We hope to receive a positive evaluation on the revised version.

Point 3: Experimental design: Page 3, line 103-110. It was not clear how many fish in each group (Pk, Bioavailability and Tissue depletion) and also for 10 mg/ kg dose and 20mg/kg dose. I request please re-write this para for better understanding.

Response 3: This information is added to an appropriate place as followings (please refer to Materials and methods, lines 103-105, p.3): The fish were divided into three groups: pharmacokinetic, bioavailability and tissue depletion group, with 180 (IM dose of 10 and 20 mg/kg, n=90, each group), 90 (IV dose of 10 mg/kg) and 60 (IM dose of 10 mg/kg) fish in each.

Point 4: 2.4. Sample preparation and HPLS-MS/MS analysis: please correct HPLS as HPLC.

Response 4: We completely agree to your comments and corrected as such. Please see the changes and additions in section 2.4. (line 129, p.3).

Point 5: Section 2.4. How absolute quantification of drug performed in serum and tissues, did authors used internal standard method or external standard method. In Bioanalysis, internal standard methods are widely used and acceptable to compensate the sample predation and analytical errors. Please provide the details of internal standard used (if internal standard method used).

Response 5: Unfortunately, we were unable to apply the internal standards (IS). Without an IS, inconsistency occurring from matrix effects and mass spectrometer itself cannot be completely compensated. It is ideal to include an IS, particularly isotope-labelled one. To overcome this problem, it was closely checked whether there is any significant change in the linearity of standard curves of both TT solutions and spiked TT standard during runs in the beginning and at the end of analysis, too. Please see the section 2.4. (lines 161-163, p.4).

Point 6: Page 4; line 153: How many standards were included in the curve. I understand its only 3 standards (correct if I am wrong).

Response 6: We realized that our description was not correct, and removed the related statements. Authors greatly appreciate for the comment. Please refer to section 2.4.(lines 155-160, p.4) for some modifications.

Point 7: Table 1: Please include column with calculate concentration.

Response 7: We appreciated the valuable comments which made us realize what we should have done. Additional data were added in the revised manuscript, with descriptions in the Table 1 (p.5).

Point 8: Discussion: please discuss about the pk values vs dose (10 mg/kg and 20 mg/kg) (dose propionate etc).

Response 8: We appreciate this valuable comment which will improve the paper. Following sentences were added in Discussion (lines 306-310, p.10): The comparative pharmacokinetic parameters for the administered dose all showed a fast absorption rate (Tmax, 0.25 h), and when administered at 20 mg/kg rather than 10 mg/kg, the Cmax was absorbed 1.5 times higher. The AUC was shown to be concentration-dependent (123.55 vs. 246.05 at 10 and 20 mg/kg) according to the administered dose.

Reviewer 2 Report

Review Report:

Journal: Animals

Manuscript ID: animals-1347596

Title:

Pharmacokinetic-pharmacodynamic profile, bioavailability, and withdrawal time of tylosin tartrate following a single intramuscular administration in olive flounder (Paralichthys olivaceus)

Authors:

Ji-Hoon Lee , Ga Won Kim , Mun-Gyeong Kwon , Jung Soo Seo *

In this article, Seo and coworkers, have found that the fishes

Tylosin tartrate is a potent bacterial-killing agent useful against frequently occurring bacterial fish infections. They tested the effectiveness against pathogenic bacteria and the human safety of the drug for possible application to cultured olive flounder, one of the most important culture species in the far eastern Asian countries. They found that tylosin tartrate was very effective in killing the pathogenic bacteria grown in artificial culture media, and it also demonstrated that the drug reached body concentrations in olive flounder, to high enough to kill the pathogen. Additionally they determined how long one should wait until the fish clears the injected drug out and is safe for human consumption. These results pave new method and safe dose of tylosin tartrate to disease treatments useful for olive flounder farming.

Since Animals  publishes original research articles, reviews and communications that offer substantial new insight into any field of study that involves animals, including zoology, ethnozoology, animal science, animal ethics and animal welfare, this article contains results that comes within the radius of aims and scope of the this journal. The article contains results that will be beneficial for the control of animal diseases, is of high impact for their health, establishes  healthy olive flounder farming and will be useful for fish husbandry and aquaculture industry. This kind of findings of pharmacological agents for controlling bacterial infections in fish farming will reduce animal diseases and boost the economy by improving fish growth and breeding. Additionally, the article also states the withdrawal time of tylosin tartrate which is beneficial for determining how long we should wait until the fish clears the injected drug out possible for human consumption and thereby ascertains the human safety dose of this drug. This work is of high public relevance in terms of both in terms, of improving financial economy of the country via improved fish farming and also ensuring delivery of healthy olive flounders  free of any pharmacological agents (highly safe) or other fishes for human consumption.

Overall Recommendation: Accept after Revisions: The paper is in principle accepted after revision based on the reviewer’s comments.

However, there are few points that need to be addressed in the paper:

  1. Figure 1 y axis and x axis labels and numerals have to be readable. Please change the font size and non-pixelated image for clarity.
  2. Is this drug effective only against Streptococcus infections in olive flounders or other bacteria in flounders and other fishes? This will establish the wide spectrum of tylosin tartrate in terms of antibacterial agent and its effectiveness on other fishes. In case, this information is found in the literature can a section be added in the Introduction section. This will prove TT as a very important pharmacological agent in treating diseases in the aquaculture industry.
